# Multifunctional, CD44v6-Targeted ORMOSIL Nanoparticles Enhance Drugs Toxicity in Cancer Cells

**DOI:** 10.3390/nano10020298

**Published:** 2020-02-10

**Authors:** Lucía Morillas-Becerril, Elektra Peta, Luca Gabrielli, Venera Russo, Elisa Lubian, Luca Nodari, Maria Grazia Ferlin, Paolo Scrimin, Giorgio Palù, Luisa Barzon, Ignazio Castagliuolo, Fabrizio Mancin, Marta Trevisan

**Affiliations:** 1Dipartimento di Scienze Chimiche, Università di Padova, via Marzolo 1, 35131 Padova, Italy; lucia.morillasbecerril@studenti.unipd.it (L.M.-B.); luca.gabrielli@unipd.it (L.G.); elisa.lubian81@gmail.com (E.L.); paolo.scrimin@unipd.it (P.S.); 2Department of Molecular Medicine, University of Padova, via Gabelli 63, 35121 Padova, Italy; elektra.peta@unipd.it (E.P.); verarusso87@gmail.com (V.R.); giorgio.palu@unipd.it (G.P.); luisa.barzon@unipd.it (L.B.); ignazio.castagliuolo@unipd.it (I.C.); 3ICMATE-CNR, Area della Ricerca di Padova, C.so Stati Uniti 4, 35127 Padova, Italy; luca.nodari@cnr.it; 4Dipartimento di Scienze Farmaceutiche, Università di Padova, Via Marzolo 5, 35131 Padova, Italy; mariagrazia.ferlin@unipd.it

**Keywords:** silica nanoparticles, targeted delivery, CD44 receptor, hyaluronic acid, antibodies

## Abstract

Drug-loaded, PEGylated, organic-modified silica (ORMOSIL) nanoparticles prepared by microemulsion condensation of vinyltriethoxysilane (VTES) were investigated as potential nanovectors for cancer therapy. To target cancer stem cells, anti-CD44v6 antibody and hyaluronic acid (HA) were conjugated to amine-functionalized PEGylated ORMOSIL nanoparticles through thiol-maleimide and amide coupling chemistries, respectively. Specific binding and uptake of conjugated nanoparticles were studied on cells overexpressing the CD44v6 receptor. Cytotoxicity was subsequently evaluated in the same cells after the uptake of the nanoparticles. Internalization of nanocarriers loaded with the anticancer drug 3N-cyclopropylmethyl-7-phenyl-pyrrolo- quinolinone (MG2477) into cells resulted in a substantial increase of the cytotoxicity with respect to the free formulation. Targeting with anti-CD44v6 antibodies or HA yielded nanoparticles with similar effectiveness, in their optimized formulation.

## 1. Introduction

The risk of developing a cancer has been constantly increasing during the last century. Today, about 16% of global deaths is due to cancer, making it the second leading cause of death [1]. Although conventional chemotherapeutics are an essential part of anti-tumor treatments, their unspecific delivery, with the subsequent decrease in efficacy and occurrence of side-effects, limits their use [2].

Therefore, the need for strategies aimed at increasing the accumulation of the drug at the target site increased the interest on nanoparticles [3,4]. These have a relevant potential as carriers for chemotherapy drugs, as well as for gene delivery and other therapeutic applications [5,6,7]. In particular, the possibility to exploit passive or active targeting may produce an enhanced accumulation in the target tissue. The most common passive targeting mechanism is based on the enhanced permeability and retention effect (EPR). Immature vascularization of tumors results in large fenestrations of the blood vessels that are not present in normal capillaries and allow nanoparticle extravasation [8]. In addition, this increased capillary permeability is accompanied by an inefficient lymphatic draining, resulting in the accumulation of nanoparticles and macromolecules in tumor tissues [9]. Passive targeting by EPR effect, however, provides results mainly on solid tumors and relies on the development of the mentioned specific physiological features, which may be different from patient to patient [9]. Similar limitations are suffered also by other active targeting mechanisms, as those based on the acidification of the tumor microenvironment [10] or the release of proteinases in the extracellular matrix [11]. 

On the other hand, active targeting consists in conjugating nanoparticles with ligands with high binding affinity for specific antigens overexpressed by cancer cells. In several studies this appears to be a more effective approach to ensure or enhance the selective accumulation of the nanosystems in the target tissue [12,13,14,15,16,17,18,19,20]. In addition, active targeting is not limited by the development of specific physiological features, since it depends only on the presence of the targeted antigen on the cancer cells. 

An attractive target for cancer therapy is represented by cancer stem cells (CSCs). CSCs are tumor cells that have the principal properties of self-renewal, clonal tumor initiation capacity, and clonal long-term repopulation potential [21]. Another peculiarity of the CSCs is the ability to evade cell death and metastasize, although they may stay dormant for long periods of time [22]. The origin of cancer stem cells remains difficult to track down. They might be derived from tissue-specific stem cells, might be initiated as a result of cell fusion or horizontal gene-transfer processes, and might also be derived from somatic cells that undergo trans-differentiation processes [23]. Because of their peculiarities, CSCs survive many commonly employed cancer therapies [24]. Consequently, a specific targeting of CSCs could be crucial in addressing this problem.

Among ligands overexpressed by CSCs, the CD44 receptor is a cell surface glycoprotein involved in several processes, including cell–cell interactions, cell adhesion, and migration [25,26,27,28,29]. As such, it plays several roles in tumor proliferation and diffusion. Several works suggested that CD44 plays a pivotal role in CSCs in communicating with the microenvironment and regulating CSC stemness properties. Moreover, CD44 is reported as the most common CSC surface marker, and, more specifically, CD44v is a promising biomarker and therapeutic target for many cancers [30]. The standard isoform has seven extracellular domains, a transmembrane, and a cytoplasmic domain and varies in size due to N- and O-glycosylation and insertion of alternatively spliced exon products. Different combinations of 1–10 variant exon products (CD44v) are inserted by alternative splicing between exons 5 and 624, and the resulting isoforms have been suggested to be relevant markers for CSCs and critical regulators of cancer stemness [31]. One of the most known markers of CSCs is the CD44v6 isoform, and its metastasis promoting activity was first described in 1991 [32]. The CD44v6 isoform is overexpressed in several CSCs (e.g., in colorectal and pancreatic cancers [33], gastric cancer [34], and ovarian cancer [35]), thus representing an ideal candidate for targeted cancer therapies.

Moreover, CD44 is the major hyaluronan receptor [36,37]. Hyaluronic acid (HA) is a natural, biodegradable, non-immunogenic, and water-soluble polysaccharide present in the extracellular matrix and synovial fluids [38,39,40,41]. Its interaction with the CD44 receptor and other receptors plays relevant roles in cancer development and metastasis dissemination [36]. In CSCs, HA supports the CD44v6 receptor activities at multiple levels [42]. In fact, binding of HA to CD44v6 facilitates tumor cell migration, receptor cross-linking, and binding of signaling molecules [43,44]. Moreover, HA contributes to inflammatory response induction of the CSCs and affects their metabolism [45]. Low molecular weight HA has been shown to promote cell motility, CD44 cleavage, and angiogenesis [28]. This interaction with cancer cells has prompted the conjugation of this biopolymer with several drug-loaded nanoparticles for its use as targeting moiety [25].

In this work, we selected organic-modified silica (ORMOSIL) nanoparticles as vectors for CD44 targeting. Silica nanoparticles have been among the most studied nanomaterials for biomedical applications in the last decade [46,47,48]. This is due to the possibility to obtain topologically precise and functionally complex structures by simple synthetic protocols [49,50,51,52]. In addition, these inorganic or hybrid nanoparticles have a highly crosslinked polymeric structure that allows the inclusion or covalent grafting of organic molecules in their interior, on the surface or within pores [53,54,55]. In the past few years, we developed [56] a synthetic protocol for the preparation of multifunctional ORMOSIL nanoparticles [56] with a one-pot reaction (Figure 1). The procedure is a modification of the original protocol proposed in 2003 by Prasad and co-workers [57], and involves the base-catalyzed condensation of vinyltriethoxysilane, VTES, in an aqueous solution containing a surfactant. Molecules, such as coating agents, dyes, photosensitizers or drugs, can be entrapped or chemically conjugated to specific regions of the nanoparticles as a result of their hydrophobic/hydrophilic balance [50,51,52,58,59]. In particular, we could prepare with this approach densely PEGylated drug-loaded nanoparticles conjugated with targeting agents [49,50].

Among anticancer agents, we selected a derivative of 7-phenylpyrrolo-quinolinone, which had previously demonstrated a significant cytotoxic activity in several cancer cell lines by acting as tubulin polymerization inhibitor [60]. Moreover, compounds of this class were shown to overcome the cross resistance observed with vincristine and paclitaxel [61,62]. However, these derivatives, in particular the 3N-cyclopropylmethyl-7-phenyl-pyrrolo-quinolinone (MG2477) required prolonged incubation times (48 to 72 h) to produce relevant effects [61]. This could be related to a slow uptake [61]. Hence, the use of tailored nanocarriers could significantly improve the effectiveness of these molecules.

Herein, we report the design, synthesis and investigation of a CD44v6-targeting nanocarrier system based on multifunctional self-organized ORMOSIL nanoparticles. The nanoparticles were loaded with the MG2477 drug or with Rhodamine B silanized derivative, coated with a dense PEG_2000_ layer, and conjugated with an anti-CD44v6 monoclonal antibody or with HA. Results demonstrated that these nanocarriers were selectively internalized by cells overexpressing the CD44v6 and significantly increase the cytotoxicity of the transported drug upon short incubation times.

## 2. Materials and Methods 

### 2.1. General 

2-(4-Chlorosulfonylphenyl)ethyltrimethoxysilane (50% solution in dichloromethane) was obtained by ACROS (Thermo Fisher Scientific, Waltham, MA, USA) and used without further purification. All the others chemical reagents were bought from Merck KGaA (Darmstadt, Germany) at highest commercial quality and used without further purification. Ultrapure water was purified using a Milli-Q^®^ water purification system (Merck KGaA). Solvents were of analytical reagent grade, laboratory reagent grade or HPLC grade. Reactions were monitored by Thin Layer Chromatography (TLC) on 0.25 mm 60 F254 Merck silica gel plates (Merck KGaA) and, when necessary, stained with ninhydrin. ORMOSIL nanoparticles (NPs) and amine-functionalized ORMOSIL nanoparticles (NH_2_-NPs) were prepared according to previously reported procedures [50,51], with small modifications, and details are given in Section 2.1.3 and Section 2.1.4.

NMR spectra in the solution state were recorded on a AVIII 500 spectrometer (Bruker, Billerica, MA, USA) operating at 500 MHz for 1H. UV-Vis absorption spectra were measured in water on a Cary 50 spectrophotometer (Agilent, Santa Clara, CA, USA) with 1-cm path length quartz cuvettes. Fluorescence spectra were measured in water or Phosphate-Buffered Saline (PBS) 1 mM buffer at pH 7 on a Cary Eclipse fluorescence spectrophotometer (Agilent). Both the spectrophotometers were equipped with thermostatted cell holders. Ultrafiltration of dye-loaded, drug-loaded and conjugated nanoparticles with HA was carried out using a disposable 15 mL centrifuge tubes and filtration through 0.45 µm cellulose acetate membrane. Ultrafiltration of nanoparticles conjugated with antibody was carried out using a 75 mL Amicon Ultrafiltration Cell using a cellulose membrane with a cut-off of 10 kDa (Merk, Darmstadt, Germany).

Infrared spectra were recorded on a low-e microscope slide (Kevley Technologies, Chesterland, Ohio, OH, USA). Once the solvent evaporated, the spectra were collected by means of a Nicolet is 10 spectrometer (Thermo Fisher Scientific, Waltham, MA, USA) coupled to a Spectra Tech Continuum optical microscope (Spectra Tech Inc., Oak Ridge, TN, USA). The spectra (64 scans) were acquired in reflection mode, in the 4000–650 cm^−1^ range, with a resolution of 4 cm^−1^.

The hydrodynamic particle size (Dynamic Light Scattering, DLS) and Z-potential were measured with a Zetasizer Nano-S (Malvern, Grovewood Road, UK) equipped with an HeNe laser (633 nm) and a Peltier thermostatic system. Measurements were performed at 25 °C in water or PBS buffer at pH 7.0. Transmission electron microscopy (TEM) images were recorded on a FEI Tecnai G12 microscope (Thermo Fisher Scientific) operating at 100 kV. The images were registered with a OSIS Veleta 4K camera (EMSIS GmbH, Munster, Germany). Thermogravimetric analysis (TGA) was run on 100 µL nanoparticle samples using a Q5000 IR instrument (TA Instruments, New Castle, DE, USA) from 25 to 1000 °C under a continuous air flow.

#### 2.1.1. Synthesis of PEG_2000_-OMe-Si (2)

PEG_2000_-OMe-Si was prepared as previously reported [49,50]. In brief, O-2-Aminoethyl)-O′-methylpolyethylene glycol (PEG_2000_-amine, 250 mg, 0.125 mmol) was dissolved in 5 mL of dry dichloromethane, followed by addition of Et_3_N (61 µL, 0.438 mmol) and 164.5 µL of 2-(4-chlorosulfonylphenyl) ethyltrimethoxysilane (50% solution in dichloromethane, 0.25 mmol). The mixture was stirred for 4 h at 42 °C in a Schlenk tube under N_2_ atmosphere. The solvent was evaporated, and the residue was dissolved in 0.3 mL ethanol. The product was precipitated by adding 13 mL of cold tert-butyl methyl ether and centrifuged at 4 °C. The purification step was repeated three times, and the white precipitate was dried under vacuum (70%). 1H NMR (500 MHz, CDCl3) δ: 1.00 (m, 2H), 2.82 (m, 2H), 3.11 (m, 2H), 3.38 (s, 2H), 3.64 (m, ~180H), 5.47 (m, 1H), 7.33 (d, J = 8.25 Hz, 2H), 7.77 (d = 8.25 Hz, 2H).

#### 2.1.2. Synthesis of PEG_3000_-NH_2_-Si (3)

PEG_3000_-NH_2_-Si was prepared as previously reported [49,50]. In brief, O,O′-Bis(2-aminoethyl)polyethylene glycol (PEG_3000_-amine, 20 mg, 0.0066 mmol) was dissolved in 400 µL of dry DMSO, followed by addition of 4.3 µL of 2-(4-chlorosulfonylphenyl) ethyltrimethoxysilane (50% solution in dichloromethane, 0.0066 mmol). The mixture was stirred for 2 h at 42 °C in a Schlenk tube under N_2_ atmosphere. The product was directly used for nanoparticles synthesis.

#### 2.1.3. Synthesis and Purification of Rhod-NPs and NH_2_-Rho-NPs

PEG_2000_-OMe-Si (30 mg, 0.013 mmol) was dissolved in 4.16 mL of ultrapure water in a thermostatted reaction vessel at 30 °C. Then, 833 µL of Brij35P water solution (30 mM, pH 2), 150 µL of n-butanol, rhodamine-B silane (25 µL, 2.5 × 10^−4^ mmol), and vinyl-triethoxysilane (VTES, 100 µL, 95 mM) were added. The mixture was stirred vigorously for 30 min, followed by addition of PEG_3000_-NH_2_-Si (70 µL, 0.011 mmol), when required, and 10 µL aqueous ammonium hydroxide (1:1 solution). After stirring for 2 h at 30 °C, the suspension was filtered through a 0.45 µm membrane filter, afterwards nanoparticles were pelleted by centrifugation and resuspended with 3 × 5 mL ultrapure water (23,500× *g*, 30 min, 3 times) and stored at 4 °C.

#### 2.1.4. Synthesis and Purification of MG2477-NPs and NH_2_-MG2477-NPs

PEG_2000_-OMe-Si (30 mg, 0.013 mmol) was dissolved in 4.16 mL of ultrapure water in a thermostatted reaction vessel at 30 °C. Then, 833 µL of Brij35P water solution (30 mM, pH 2), 150 µL of n-butanol, drug (150 μL, 0.0015 mmol), and vinyl-triethoxysilane (VTES, 100 μL, 95 mM) were added. The mixture was stirred vigorously for 30 min, followed by addition of PEG_3000_-NH_2_-Si (70 µL, 0.011 mmol), when required, and 10 µL aqueous ammonium hydroxide (1:1 solution). After stirring for 2 h at 30 °C, the suspension was filtered through a 0.45 µm membrane filter; afterwards, nanoparticles were pelleted by centrifugation and resuspended with 3 × 5 mL ultrapure water (23,500× *g*, 30 min, 3 times) and stored at 4 °C.

#### 2.1.5. Conjugation of Anti-CD44v6 Antibody to NPs (Ab-CD44v6-NPs)

For the derivatization of anti-CD44v6 antibodies (Ab-CD44v6), 280 μL of 3.46 mg/mL Ab solution (obtained as described in Section 2.2.2) in 10 mM PBS and 28 μL of 1M NaHCO_3_ were mixed with 0.013 μL of 0.1 M ethylenediaminetetraacetic acid (EDTA, pH = 8.00). Under these conditions, the concentration of EDTA in the final mixture was 4 mM. Subsequently, 1.94 μL of 2-iminothiolane in ultrapure water was added, having a ratio 1:7 Ab/2-IT. The reaction mixture was incubated at room temperature (RT) for 2 h and then at 4 °C overnight. Afterwards, 30 μL of 2 M glycine was added and the mixture was stirred for 20 min at RT. The derivatized antibodies were purified by ultrafiltration with Amicon Ultra (cut-off 10 kDa) with PBS/EDTA 1 mM/4 mM (23,500 rpm, 10 min, 4 times) to a final volume of 280 μL.

The activation of NH_2_-NPs was carried out by reacting 550 μL of 2.46 mg/mL NH_2_-NPs in 10 mM PBS with an excess of 3-(maleimido)propionic acid N-hydroxysuccinimide ester (1 mg) and 60 μL of 1 mM NaHCO_3_. The reaction mixture was incubated at 30 °C overnight. The activated nanoparticles were purified by ultrafiltration with Amicon Ultra (cut-off 10 kDa) with PBS:EDTA 1 mM/4 mM (9180 g, 20 min, 3 times) to a final volume of 550 μL.

Subsequently, the activated nanoparticles were mixed with 28 μL of iminothiolane derivatized Ab-CD44v6 to obtain the ratio 1×, and with 280 μL of Ab-CD44v6 for the ratio 10×. The reaction mixture was stirred at RT for 2 h and then at 4 °C for 48 h. Ab-CD44v6-NPs were purified by ultrafiltration with Amicon Ultra 3 kDa with PBS:EDTA 1 mM/4 mM (23,500× *g*, 10 min, 4 times) and stored at 4 °C.

#### 2.1.6. Conjugation of HA to NPs

N-(3-Dimethylaminopropyl)-N′-ethylcarbodiimide hydrochloride (for 11.5 kDa HA, 30.8 mg, 0.16 mmol; for 22.5 kDa HA, 15.4 mg, 0.08 mmol) and N-hydroxysuccinimide (for 11.5 kDa HA, 9.2 mg, 0.0.08 mmol; for 22.5 kDa HA, 4.6 mg, 0.04 mmol) in 1 mL ultrapure water were added to a solution of HA (11.5 kDa HA, 8.8 mg, 7.8 × 10^−4^ mmol. 22.5 kDa HA, 17.6 mg, 7.8 × 10^−4^ mmol) in HEPES buffer (1 mL, 10 mM, pH 6). The mixture was stirred for 30 min for the activation of HA (1.8 mL, 1× conjugation; 0.2 mL, 10× conjugation), and then 2 mL of 4.29 mg/mL NH_2_-NPs were added to each reaction vessel. After overnight stirring, the nanoparticles were filtered through a 0.45 µm membrane filter, pelleted by centrifugation, and resuspended with 5 mL ultrapure water (23,500× *g*, 30 min, 3 times) and then with PBS 1 mM (23,500× *g*, 30 min) and stored at 4 °C.

#### 2.1.7. Quantification of Antibody Conjugated to Ab-CD44v6-NPs

The concentration of the antibody conjugated to the NPs was analyzed by Sodium Dodecyl Sulphate-PolyAcrylamide Gel Electrophoresis (SDS-PAGE). Briefly, the antibody was denatured (99 °C for 5 min and β-mercaptoethanol), loaded on a 10% polyacrylamide gel and run at 70 mA. The gel was then colored with Brilliant Blue (Merck) and washed with an aqueous solution composed of 10% glacial acetic acid and 45% methanol (both from Merck). Samples with different known concentrations of γ-globulins or Ab-CD44v6 were used to set up a calibration curve. Quantification was performed by plotting the bands density and extrapolating the final concentration form the standard curve. The concentration of antibody in the drug-loaded NPs conjugated with the antibody in ratio 1:1 was 94 ng/μL; meanwhile, the drug-loaded NPs conjugated with the antibody in ratio 10:1 was 24 ng/μL.

### 2.2. Cellular Uptake and Cytotoxicity 

#### 2.2.1. Cell Cultures 

HEK-293A and HeLa cells were grown in Dulbecco’s Modified Eagle’s Medium (DMEM) supplemented with 10% fetal bovine serum (FBS), 1% GlutaMAX^TM^ and 1% penicillin/streptomycin (all from Thermo Fisher Scientific). The cells were maintained in a humidified incubator at 37 °C in an atmosphere composed of 5% CO_2_. Hybridoma cell line GK1.5 (ATTC, Manassa, VA, USA) were grown in suspension in DMEM supplemented with 20 FBS, 1% GlutaMAX^TM^, 1% penicillin/streptomycin and 1% pyruvic acid (Thermo Fisher Scientific). 

#### 2.2.2. Production and Purification of Anti-CD44v6 Antibody

To produce the antibody Ab-CD44v6, hybridoma cells GK1.5 were grown for 10 days in DMEM without FBS. After 10 days, cells were centrifuged at 250× *g* for 10 min and cell supernatants were collected and filtered through a 0.22 µm filter. Proteins present in the supernatant were then concentrated by using an Amicon Ultra Filter with a cut-off of 30 kDa. The antibody anti-CD44v6 was purified by Montage Antibody Purification with PROSEP^®^-G (Merck) following manufacturer instructions. Concentration of isolated antibody was determined by spectrophotometer analysis with a Nanodrop instrument (Thermo Fisher Scientific) and verified by SDS-PAGE as described in Section 2.1.7.

#### 2.2.3. Transfection of Cell Cultures

HEK-293A-CD44v6 and HeLa-CD44v6 were obtained by transfection, with Lipofectamine^®^2000 (Invitrogen, Thermo Fisher Scientific) of a pCMV6Entry (OriGene Technology, Rockville, MD, USA, CAT#: PS100001) harboring the CD44v6 cDNA under the promoter of human cytomegalovirus CMV). Briefly, 10^5^ cells were seeded in 6-well plates, and, once 70–80% of confluency was reached, cells were transfected with 2.5 µg of plasmid DNA. Twenty-four hours later, G418 (Thermo Fisher Scientific) was added to cells at a concentration of 800 µg/mL and maintained until the selection was over. Analysis of CD44v6 expression was performed by RT-PCR analysis. RNAs were isolated from cells with RNeasy Mini Kit (Qiagen, Hilden, Germany) and used for reverse transcription. A RT-PCR was applied to amplify the region specific for CD44v6 using the primers Fw: CATCTACCCCAGCAACCCTA, Rw: TGGGTCTCTTCTTCCACCTG with the following conditions: 95 °C 10 min, 35 cycles: 95 °C 30 s, 57 °C 30 s, 72 °C 30 s; 72 °C 7 min. Amplicons were subsequently loaded into a 1.5% agarose gel for electrophoresis run (Appendix A).

#### 2.2.4. Binding and Internalization Experiments

For binding and internalization experiments, cells were seeded at the concentration of 20,000 cells per cm^2^ in 24-well plates and let to adhere for 24 h at 37 °C. For binding, cells were fixed in Paraformaldehyde (PFA) 4% for 20 min at RT and subsequently treated with different concentrations of NPs at RT for 1 h. For internalization, cells were treated with NPs at different concentrations for 4 h at 37 °C. Cells were washed multiple times to eliminate unbound NPs and fixed in PFA 4% for 20 min at RT and observed at a Leica DFC420 inverted epifluorescence microscope (Leica Biosystems, Wetzlar, Germany). For fluorescence-activated cells sorting (FACS) analysis: internalized cells were evaluated by detaching the cells with trypsin, washing them with PSB, and by analyzing the fluorescent signal with a BD FACSCalibur (BD Bioscience, Franklin Lakes, NJ, USA).

#### 2.2.5. Confocal Experiments

For Zeta stack analysis, cells were seeded on glass coverslips in 24-well plates and allowed to adhere for 24 h. The next day, cells were washed with PBS and a solution of Carboxyfluorescein succinimidyl ester (CFSE) 10 µM CellTrace (Thermo Fisher Scientific) was added and incubated for 20 min at 37 °C in the dark. Cells were then washed with PBS to remove excess of CSFE, and they were let to recover for 1 h at 37 °C in growth medium. Cells were then incubated with NPs for 4 h at 37 °C and fixed, as previously described, and observed at a confocal scanner laser microscopy (CLSM, Nikon Eclipse Ti, Nikon, Minato, Tokyo, Japan).

#### 2.2.6. MTT Cell Viability Assay

Cell viability was evaluated by MTT colorimetric assay (Merck). Cells were seeded in 96-well tissue plates and allowed to adhere overnight. The day after, cells were exposed to NPs for 4 h at 37 °C in the presence of DMEM medium supplemented with 2% FBS. Cells were then washed three times with DMEM without FBS to eliminate the unbounded NPs and were allowed to further grow for 72 h in DMEM supplemented with 10% FBS, 1% GlutaMAX, and 1% penicillin/streptomycin. Subsequently, 10 μL a solution of freshly dissolved MTT (5 mg/mL in PBS) were added to each well, and, after 4 h of incubation, 100 μL of solubilization solution (10% sodium dodecyl sulfate (SDS) and 0.01 M HCl) were added. The colorimetric reaction was spectrophotometer at a wavelength of 620 nm, using an absorbance reader for 96-well plates (Sunrise, Tecan, Switzerland).

#### 2.2.7. Competition Assay

Hela-CD44v6 cells were seeded in 96-well dishes. The day after, cells were incubated in excess of free HA (10 mg/mL, Lifecore Biomedical, Chaska, MN, USA) in DMEM medium without FBS for one h. Control cells were treated with the same medium without the presence of HA. Subsequently, cells were washed and incubated for 4 h at 37 °C with HA-conjugated and -unconjugated NPs (0.1 μM). To evaluate cell cytotoxicity, MTT assay was performed 72 h post-treatment. The experiment was carried out in triplicate.

## 3. Results and Discussion

### 3.1. Synthesis and Conjugation of the Nanoparticles

The synthesis of PEGylated ORMOSIL nanoparticles (NPs) is summarized in Figure 1 [49,50]. The process requires the polymerization of vinyltriethoxysilane catalyzed by ammonia as base, in an aqueous solution containing the surfactant Brij 35. The surfactant favors the pre-organization of the NP components and monomers, which locate themselves in different NP regions depending on their solvophilicity. Hydrophobic species, such as dyes, drugs, and VTES, locate in the NPs core, where the polymerization reaction condense them to form the ORMOSIL matrix. Organic species result entrapped or chemically grafted to the ORMOSIL network, depending on the presence or absence of reactive silane groups in their structure. Surfactants and amphiphilic species locate on the NPs surface, interacting with both the core and the surrounding solution.

PEG derivatives PEG-Si and NH_2_-PEG-Si (Figure 1) used to coat the ORMOSIL NPs were obtained by a Hinsberg reaction of the corresponding PEG amine with 2-(4-chlorosulfonylphenyl) ethyltrimethoxysilane (Figure 2) [49,50]. Products can be used without purification or after precipitation with tert-buthyl methyl ether. In the case of NH_2_-PEG-Si, a small fraction of bi-functionalized derivative (bearing two silane moieties) can be present, but this does not affect the properties of the nanoparticles [49,50].

As previously reported, the size of NPs is controlled by the concentration of VTES in the reaction mixture [49]. We chose a 95 mM concentration, which was expected to provide NPs with a diameter of about 120–140 nm. To obtain NH_2_-NPs, PEG-Si and NH_2_-PEG-Si were added in a 10:1 ratio at the same total concentration of the Brij 35 surfactant (5 mM). Derivative NH_2_-PEG-Si ensures the presence in the coating of the primary amino groups needed for the conjugation with the targeting agents. The presence of the amino groups was confirmed with the fluorescamine test (Appendix A). The cytotoxic drug MG2477 (0.3 mM) or the Rhodamine B fluorophore (Rho; 0.05 mM) were loaded into NPs and NH_2_-NPs simply by adding them to the reaction mixture. It must be noted that, while Rho copolymerize with VTES resulting in a covalent grafting, MG2477 remains simply entrapped in the hydrophobic ORMOSIL matrix and its release is possible [63].

The conjugation of the NH_2_-NPs with the anti-CD44v6 antibody (Ab-CD44v6) was obtained by the thiol-maleimide chemistry, introducing maleimide moieties in the nanoparticles and thiol groups in the antibody (Figure 3). The antibody was reacted with 2-iminothiolane (2-IT) in a 1:7 Ab-CD44v6/2-IT ratio to introduce about 2-4 sulfhydryl groups per protein. Functionalization was confirmed by the Ellman test [64,65]. NH_2_-NPs nanoparticles were reacted with 3-(maleimido) propionic acid N-hydroxysuccinimide ester (MBS). Purification was performed by ultrafiltration with membranes with a 10 kDa cut-off. The successful conversion of the amino groups of NH_2_-NPs in maleimide groups was verified with the fluorescamine test that confirmed the decrease of the amount of NH_2_ groups in the nanoparticles’ coating. The conjugation of the maleimide-NPs with the antibody was performed by incubating them in different ratios at RT for 48 h in PBS. Two different batches of conjugated NPs were prepared using 10-fold different amounts of antibody (1× and 10×, Section 2.1.5, Appendix A). The so-obtained Ab-CD44v6-NPs were purified by ultrafiltration. As control, conjugation of human serum albumin (Alb) with NH_2_-NPs was performed. Alb was functionalized by reaction with 2-IT, followed by the conjugation with maleimide-NPs under the same conditions as the antibody conjugation.

NH_2_-NPs were also conjugated with HA fragments with different molecular weights (11.5 and 22.5 kDa) and different ratios (Figure 4). Some of the carboxylic groups of HA were converted in active esters by reaction with N-hydroxysuccinimide (NHS) and N-(2-dimethyl-aminopropyl)- N’-ethylcarbodiimide (EDC) as coupling agent. The amine-bearing NH_2_-NPs were then added and incubated overnight at RT. Also in this case, different batches of conjugated NPs were prepared using 10-fold different amounts of HAs (1× and 10×, Section 2.1.6, Appendix A). The HA-NPs were purified by centrifugation and stored in PBS 1 mM solution.

### 3.2. Characterization of the Conjugated Nanoparticles

The NPs were characterized by dynamic light scattering (DLS), zeta potential, fluorescence emission spectroscopy, thermogravimetric analysis (TGA), and transmission electron microscopy (TEM) (Appendix A). DLS (Table 1) and TEM (Figure 5) analyses confirmed that the average diameter of the NPs before conjugation ranged from 100 to 140 nm, depending on the preparation (entries 1 and 4). Polidispersity index (PDI) values indicate a homogenous size distribution as confirmed by TEM analysis. 

The Z-potential of the NPs before conjugation (in PBS buffer, pH 7) was −1 mV (Table 1). Z-Potential measures the electric potential of the double layer at the location of the slipping plane and provides information about the surface charge of a colloid. As we already reported [27], NPs and NH_2_-NPs have slightly negative Z-potential, suggesting that most of the surface is coated by PEG or vinyl groups, with only a few residual deprotonated silanol groups responsible for the small negative charge.

The concentration of the anti-cancer drug MG2477 (Figure 1) loaded into NH_2_-NPs was calculated by fluorescence spectroscopy using a calibration curve built in a solvent mixture which could reproduce the position of the emission maximum, hence matching the internal polarity of the NPs. A concentration of 46.5 μM was obtained for a 4.30 mg/mL NH_2_-NPs suspension, corresponding to a 0.4% w/w loading.

Conjugation with Ab-CD44v6 resulted only in a slight decrease of the NPs Z-potential, as expected on the basis of the typical isoelectric value of antibodies, which is close to neutrality. The average diameter measured with DLS remained almost unaffected, but the PDI values underwent a substantial increase, indicating the formation of a few aggregates in the sample. Indeed, after prolonged conservation of the samples at 4 °C the formation of sediments, which could be readily re-dispersed by sonication, was observed. TEM images confirmed, however, that the ORMOSIL cores remained unaffected and aggregation was not extensive.

Protein conjugation could be demonstrated by SDS-PAGE analysis in reducing conditions, i.e., treatment with heat and β-mercaptoethanol. This caused the detachment of part of the antibody from the NPs, due to the breaking of the disulfide bridges, allowing the antibody to run along the polyacrylamide gel (Figure 6). In the absence of reducing agents, neither free Ab-CD44v6 nor free Alb were detected in the electrophoretic run. However, in reducing conditions, bands corresponding, respectively, to the detached antibody heavy and light chains could be observed. This confirmed that proteins were covalently bound to the NPs and could be detected only when the maleimide-thiol bond was cleaved [66].

Titration of conjugated antibody was performed as described in Section 2.1.7 by SDS-PAGE by loading into a polyacrylamide gel the sample of Ab-CD44v6-Rho-NPs together with different known concentrations of Ab-CD44v6 and by extrapolating the concentration by comparison with the calibration curve (Appendix A). The concentration of antibody in 1 mg/mL of Rho loaded NPs, conjugated with the lower amount of antibody (hereinafter referred to as Ab-CD44v6^1x^-Rho-NPs), was 0.03 mg/mL. This corresponded to a 3% w/w ratio. Approximate geometrical calculations indicated that about 15% of the particles surface was coated by the antibodies (~120 antibody molecules per NP). In the case of a 1 mg/mL suspension of MG2477-loaded NPs conjugated with the antibody at the highest amount (hereinafter referred to as Ab-CD44v6^10x^-MG2477-NPs), a 0.162 mg/mL concentration of antibody was obtained. This corresponded to a 16% w/w ratio (~640 antibody molecules per NP) and an almost total surface coverage (86%).

In the case of HA-NPs, the conjugation had a more relevant effect on the Z-potential value, which decreased from −1 mV to −7/−8 mV (Table 1). Such an effect is in agreement with the polyanionic nature of the polymer. The average diameter measured with DLS, as well as the PDI, remained almost unaffected, and TEM images showed unaltered ORMOSIL cores (Figure 5D). In agreement, no tendency to sedimentation was observed. Micro-FT-IR experiments confirmed the presence of HA in the purified HA-NPs samples (Figure 7). The μ-FT-IR spectra of the NPs contained several signals, among which it was possible to easily detect those arising, respectively, from the CH and NH stretching (3400–2900 cm^−1^), the NH bending (1600–1400 cm^−1^), CC and CO stretching (1200–1000 cm^−1^), and SiO stretching (1000–800 cm^−1^). After conjugation with HA fragments, the spectrum was very close to that of the unconjugated particles but for a few signals in the 1800–1200 cm^−1^ regions, where C=O stretching signals from carboxyl (1743 cm^−1^) and amide residues (AI, 1652 cm^−1^, AII, 1543 cm^−1^) were detected. Noticeably, the presence of the amide signals clearly confirmed the effective conjugation of the biopolymer to the NPs (Figure 7, Appendix A).

The concentration of the drug MG2477 in the NPs was monitored after conjugation with the targeting agents. In both the cases of Ab-CD44v6 and HA, a relevant decrease of drug MG2477 loading was observed, likely due to leaking of the entrapped drug during the conjugation and purifications procedures (Table 2). The residual amount of encapsulated drug was, however, sufficient to produce relevant cytotoxicity.

### 3.3. Nanoparticles Interaction with Cells

#### 3.3.1. Cell Attachment and Internalization of Ab-CD44v6-Rho-NPs

NPs conjugated with the smaller amounts of targeting agents (1×) were used for the in vitro experiments when not otherwise stated. Binding of NPs to cells was analyzed by incubating increasing concentrations of NPs, loaded with Rho, and conjugated with Ab-CD44v6 (Ab-CD44v6-Rho-NPs), with HEK-293A cells overexpressing CD44v6 (HEK-293A-CD44v6), at RT for 1 h, to allow in such a short incubation time only the binding of NPs to cells without internalization. After washing out unbounded NPs multiple times and fixing cells, Rho signal of bonded NPs was analyzed at a fluorescence microscope (Figure 8A). Fluorescence intensity of Rho increased in an NP-concentration-dependent manner, indicating that Ab-CD44v6-Rho-NPs were attached to cells. In contrast, control NPs and NPs loaded with Rho and conjugated to human albumin (Alb-Rho-NPs) did not show any specific signal, even when used at the maximum concentration (Figure 8A and Appendix A).

Selective internalization and intracellular distribution of NPs in HEK-293A-CD44v6 cells was analyzed by incubating the cells with different concentrations of Ab-CD44v6-Rho-NPs, Alb-Rho-NPs and unconjugated NPs (Rho-NPs), at 37 °C for 4 h, to allow entry of NPs in cells. Cells were then washed multiple times in order to remove any unbound NP and Rho signal was analyzed by fluorescence and confocal microscopy. The Rho signal was distributed in both nuclei and cytoplasm and was dose-dependent (Figure 8B). On the contrary, no fluorescence was detected in cells incubated with the same concentrations of Alb-Rho-NPs or Rho-NPs (Figure 8B), suggesting an Ab-CD44v6-dependent uptake mechanism. Accordingly, fluorescence-activated cells sorting (FACS) analysis in similar experiments with higher concentrations of NPs on HEK-293A-CD44v6 cells confirmed the dose-dependent internalization of Ab-CD44v6-Rho-NPs (Figure 8C,D). Up to 85% of Rho positive cells were found at the highest concentration of 1 mg/mL, without affecting cells survival (data not shown). Confocal laser scanning microscopy (CLSM) was used to confirm the effective internalization of Ab-CD44v6-Rho-NPs in HEK293A-CD44v6 cells. A Z-stack analysis revealed the localization of conjugated NPs within cell cytoplasm (colored in green, Figure 8E), as observed in the sagittal (YZ) and frontal (XY) planes of the section, while no Rho signal was observed in unconjugated Rho-NPs used at the same concentration (Figure 8E), thus corroborating the receptor-mediated specific internalization of conjugated NPs. 

Additional control experiments, performed using both CT-26 cells (expressing the CD44v6) and bovine cells 5050 (not expressing the CD44v6), showed that Ab-CD44v6-Rho-NPs can be efficiently internalized only in cells expressing CD44v6 (Appendix A). 

#### 3.3.2. Cell Attachment and Internalization of HA-Rho-NPs

Similar results were obtained by using NPs conjugated with HA 22.5 kDa (HA-Rho-NPs). While unconjugated Rho-NPs did not show any internalization into cells, a strong Rho fluorescence signals was observed in samples treated with conjugated NPs, both by fluorescence (Figure 9A and Appendix A) and Z stack (Figure 9B) analysis. Lastly, uptake of conjugated NPs could not be prevented by interactions of cells with a non-specific antibody (Appendix A).

#### 3.3.3. Delivery of the Quinolone Derivative MG2477

MG2477 is a potent growth inhibitor of several cancer cell lines in vitro. It acts by inhibiting tubulin polymerization and causes cells to arrest in metaphase [38,42]. Cell viability tests after NPs administration were performed in HeLa cells engineered to overexpress CD44v6 (HeLa-CD44v6), due to their higher adhesion property to multi-wells plates, compared to HEK-293A. Cytotoxicity of unconjugated and unloaded NPs and of the free MG2477 was first investigated by (3-(4,5-dimethylthiazol-2-yl)-2,5-diphenyltetrazolium bromide (MTT) assay. Unconjugated and unloaded NPs were incubated with cells at 37 °C for 4 h, after which the cells were washed to eliminate non-internalized NPs. At 24 and 72 h post-treatment, MTT assay was performed to measure cell viability. NPs did not show any significant toxicity up to the concentration of 500 µg/mL, and for incubation times as long as 72 h, indicating the good biocompatibility of these nanocarriers (Figure 10A). To analyze the cytotoxicity of free MG2477, different concentrations of drug ranging from 0.01 to 5 µM were used to treat cells and cell survival was measured by MTT assay after different incubation times (Figure 10B). As already reported [62,63], MG2477 demonstrated a relevant cytotoxic activity. Indeed, cell viability was reduced by 20% at 0.01 µM concentration and by about 60% at concentrations higher than 0.1 µM. However, such activity could be detected only for incubation times longer than 24 h, whereas viability reduction was irrelevant, at any drug concentration, for shorter incubation times (Figure 10B). 

To investigate the effectiveness of targeted drug delivery with the NPs prepared, conjugated and unconjugated MG2477-loaded NPs, as well as the free drug, were incubated with HeLa-CD44v6 cells at the concentration of 0.1 µM (unless otherwise stated) for 4 h at 37 °C, then the cells were washed and allowed to grow for further 72 h in a renewed medium before MTT assay was applied. When Ab-CD44v6-MG2477-NPs were tested at either 0.01 or 0.1 µM (final drug concentration), a cell viability reduction in the range of 20–40% was observed (Figure 10C). 

At both concentrations, NPs conjugated with a higher amount of antibody (Ab-CD44v6^10x^-MG2477-NPs, antibody amount 10×) produced a lower cytotoxic effect compared to those conjugated with a lower amount (Ab-CD44v6^1x^-MG2477-NPs, antibody amount 1×). In the Ab-CD44v6^10x^-MG2477-NPs, the nanoparticle surface is almost completely covered by antibody molecules (see earlier) and such a crowding could reduce the accessibility to the binding sites of CD44 [67,68] reducing the antigen binding effectiveness of the nanoparticles. However, the decreased viability determined by Ab-CD44v6^1x^-MG2477-NPs was significantly greater compared to unconjugated MG2477-NPs (Figure 10C; *p* < 0.05 and *p* < 0.001 for 0.01 and 0.1 µM concentrations, respectively), confirming the receptor-mediated internalization of the NPs into target cells. No significant differences were observed by increasing the drug concentrations from 0.01 to 0.1 µM (Figure 10B), suggesting that uptake saturation is already reached at the lower concentration. 

Interestingly, no significant reduction of cell viability was observed when cells were incubated for 4 h with unconjugated MG2477-NPs neither at 0.01 µM nor at 0.1 µM, at 72 h post-treatment (Figure 10C). Hence, the cytotoxic effect of the drug encapsulated in unconjugated nanoparticles is even lower than that, already low, of the free drug. We earlier reported that cellular uptake of NPs is very low, presumably due to the stealth effect produces by the PEG coating [50]. Accordingly, in the absence of targeting agents, loading of drugs in NPs prevents its penetration in the cells. 

Cell viability was also evaluated for the NPs (loaded with MG2447 drug at 0.1 µM) conjugated with HA with two different molecular weights (11.5 and 22.5 kDa) and in two different amounts (1× and 10×) and compared to unconjugated NPs (Figure 10D). A significant viability reduction between 20% and 40% (*p* < 0.01) was observed for HA^1x^-MG2447-NPs and HA^10x^-MG2447-NPs, respectively, compared to unconjugated MG2477-NPs, confirming the specific HA-mediated internalization of NPs in target cells. Hence, a significant difference was observed between NPs conjugated with the different amounts of HA. On the opposite to what obtained with the antibody-conjugated NPs, a higher amount of conjugated HA (HA 10×) induced a higher cytotoxicity (*p* < 0.01) compared to the NPs conjugated with a lower amount. Being a linear polymer, HA likely is less sensitive to steric effects than Ab. In addition, no dependence from the molecular weight of the HA was observed, suggesting that the interaction with CD44v6 occurs with a small fragment of the biopolymer. 

To further demonstrate that cellular uptake of HA-NPs was due to HA-specific receptor interaction, competition experiments were performed in the presence of an excess of free HA. HeLa cells were incubated for 1 h at 37 °C with an excess of HA (10 mg/mL), subsequently washed, and then incubated with HA-MG2447-NPs for 4 h. At 72 h post-treatment, cells were analyzed by the MTT assay. A significant (*p* < 0.01) reduction of cytotoxicity was observed in cells treated with an excess of HA (Figure 10E), confirming the receptor-mediated internalization of the conjugated NPs. Even in this case, both NPs conjugated, respectively, with 11.5 and 22.5 kDa HA fragments demonstrated similar results confirming the absence of a significant correlation with the molecular weight of the HA polymer.

## 4. Conclusions

In summary, by using an anti-CD44v6 antibody or HA as targeting agents for a receptor overexpressed in cancer stem cells, we could demonstrate selective cytotoxicity of the drug-loaded nanocarriers. The results reported here not only describe an attractive biomedical application of ORMOSIL nanoparticles for cancer therapy but also highlight the general potential of nanoparticles in cancer therapy. Indeed, it is worth underlining that the use of targeted NPs allowed here to achieve a relevant cytotoxicity in conditions (concentration of the drug and incubation times) at which the free drug was almost ineffective. In other words, the use of these targeted nanovectors resulted in the enhancement of the activity of the drug. The reason for such an effect probably lays in the change of the drug availability. In the free from, MG2477 is poorly soluble in water and, for this reason, accumulates slowly in cells. This, in part, explains the need for long incubation times to observe relevant viability reduction. Targeted NPs provide a much more efficient uptake pathway. They encapsulate the drug, preventing its aggregation and precipitation, and rapidly deliver it in the cell by receptor mediated uptake. In agreement with this hypothesis, we observed that unconjugated NPs reduced the cytotoxicity of the drug, likely by preventing its internalization in the cells. 

The results obtained provide also relevant comparative information about the effectiveness of different targeting strategies. Indeed, similar cytotoxicity was obtained with the Ab-CD44v6 and HA-conjugated NPs, in their optimized formulation. From this perspective, the use of the polysaccharide appears more interesting. First, the conjugation protocol is simpler; there is no need to pre-functionalize both the NPs and the targeting agent. Second, the targeting agent itself is less expensive and easier to produce in large amounts. Third, greater NPs and, consequently, drug concentrations are obtained when NPs are targeted with HA with respect to the antibody.

Effectiveness of HA as a targeting agent appeared to be independent from its molecular weight, with the exception of small oligomers which proved to be ineffective (data not shown). Active targeting of the CD44v6 receptor was confirmed by the observation that a high concentration of free HA reduced the cytotoxicity. On the other hand, cytotoxicity was proportional to the amount of targeting agent conjugated to the NPs and high concentrations of the free targeting agent did not suppress the biological activity of the NPs. This suggests a relatively low affinity of HA for the CD44 receptors, which could have been enhanced by the NPs multivalency, or the need for the receptor clustering to promote the internalization, which could have been favored by the presence of a high number of targeting molecules on the NPs surface. 

In the case of the antibody-conjugated NPs, however, an inverse trend was observed, with NPs conjugated with a smaller amount of targeting agent being more effective than the ones conjugated with a larger amount of antibody. This observation is quite relevant as it confirms earlier observations that small amounts of antibodies are sufficient to obtain a good targeting and delivery efficiency [34,35]. When the density of antibody on the particles surface is increased, they may interfere with each other, decreasing their effectiveness. On the other hand, the affinity of the antibody for the CD44 receptor is likely larger than that of HA. For this reason, a small number of antibody molecules on the NPs is sufficient to achieve either effective binding or receptors clustering. 

## Figures and Tables

**Figure 1 nanomaterials-10-00298-f001:**
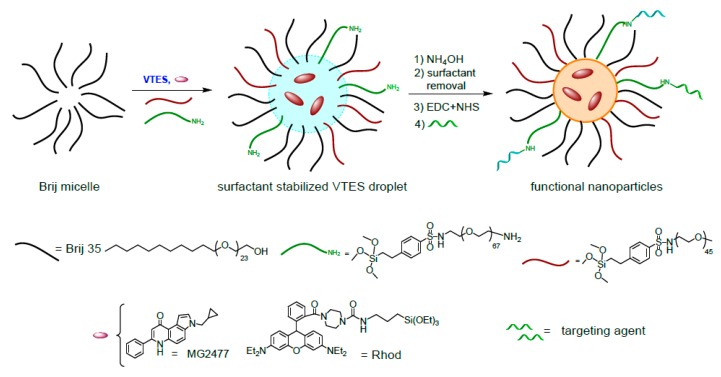
One-pot synthesis of doped, PEGylated, and functional organic-modified silica (ORMOSIL) nanoparticles (NPs). Nanoparticles will be named in the paper according to the following code: surface groups-loaded species = NP, where NP indicates a PEGylated ORMOSIL nanoparticle. VTES = vinyltriethoxysilane; NHS = N-hydroxysuccinimide.

**Figure 2 nanomaterials-10-00298-f002:**
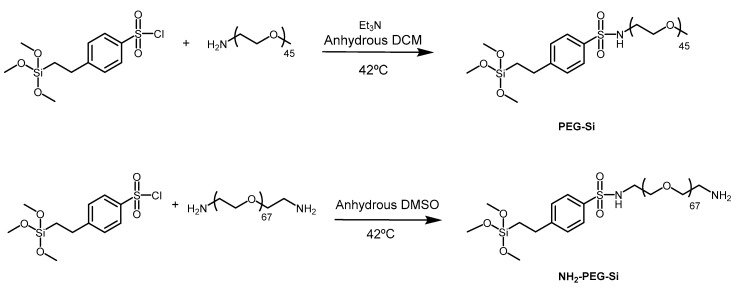
Synthesis of the PEGylated derivatives for nanoparticle coating. DCM = dichloromethane. DMSO = dimethylsulfoxide.

**Figure 3 nanomaterials-10-00298-f003:**
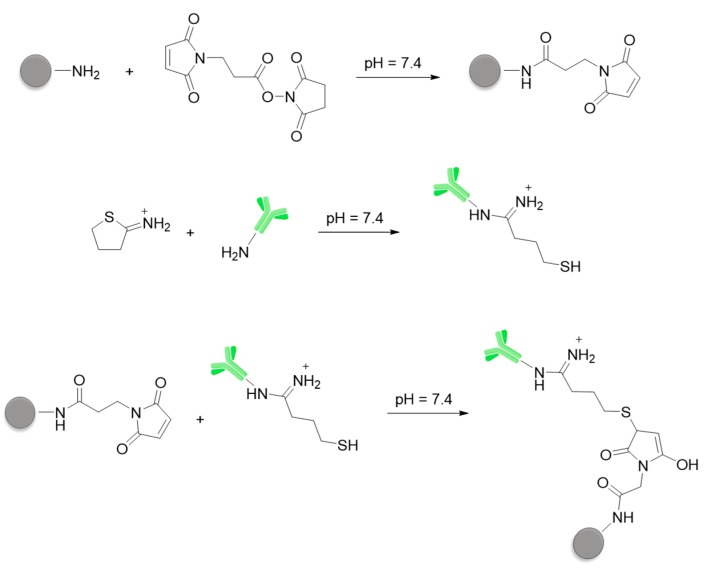
Conjugation of NH_2_-NPs with the antibody anti-CD44v6.

**Figure 4 nanomaterials-10-00298-f004:**
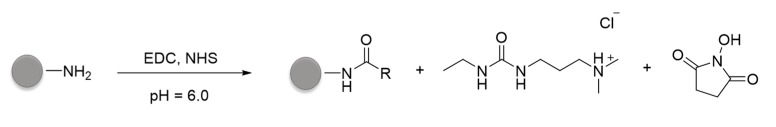
Conjugation of NH_2_-NPs with hyaluronic acid (HA).

**Figure 5 nanomaterials-10-00298-f005:**
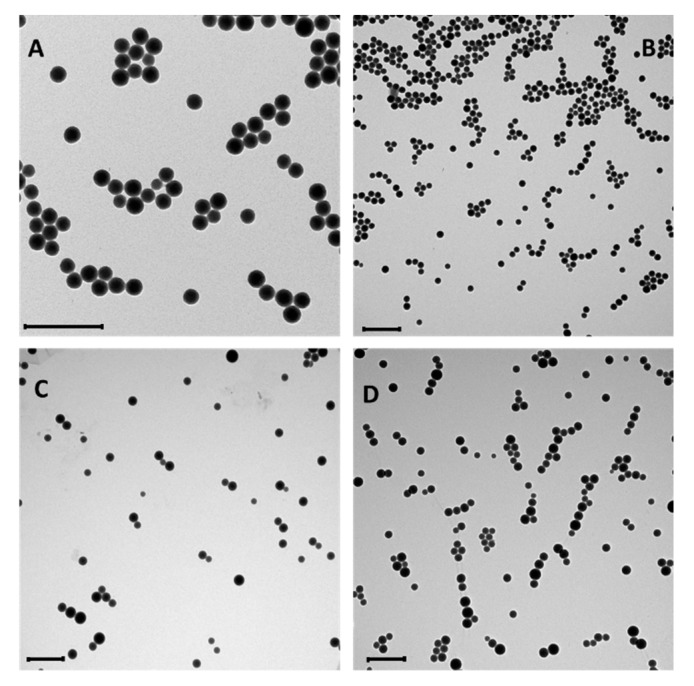
Representative TEM images of: NPs (**A**), NH_2_-NPs (**B**), Ab-CD44v6-NPs (**C**), HA-NPs (**D**). Scale bars: 500 nm.

**Figure 6 nanomaterials-10-00298-f006:**
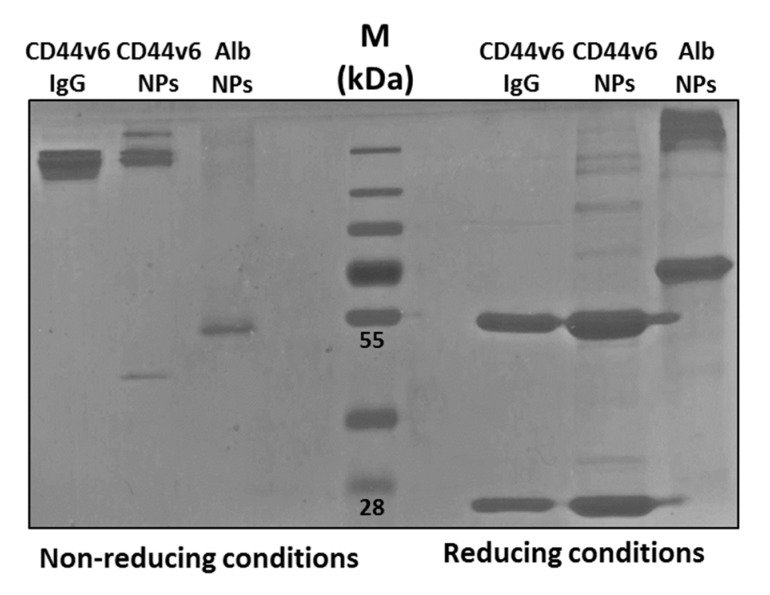
Analysis by SDS-PAGE of antibody anti-CD44v6 (ab-CD44v6) conjugation to NPs. An equal amount of NPs were treated in reducing (heat and β-mercaptoethanol) and non-reducing conditions and loaded into a 10% polyacrylamide gel. CD44v6-IgG: free Ab-CD44v6 immunoglobulin G; CD44v6-NPs: NPs conjugated with the Ab-CD44v6; Alb-NPs: NPs conjugated with Albumin; M protein Marker.

**Figure 7 nanomaterials-10-00298-f007:**
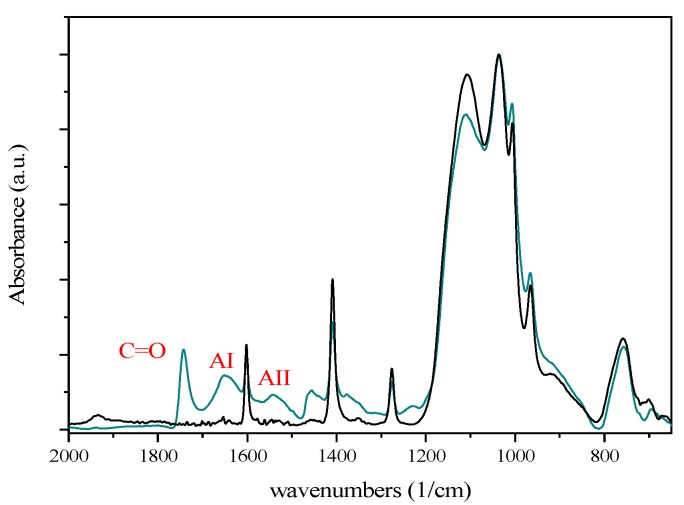
Micro-FT-IR spectra of unconjugated NH_2_-NPs (black line) and HA-NPs (dark cyan line).

**Figure 8 nanomaterials-10-00298-f008:**
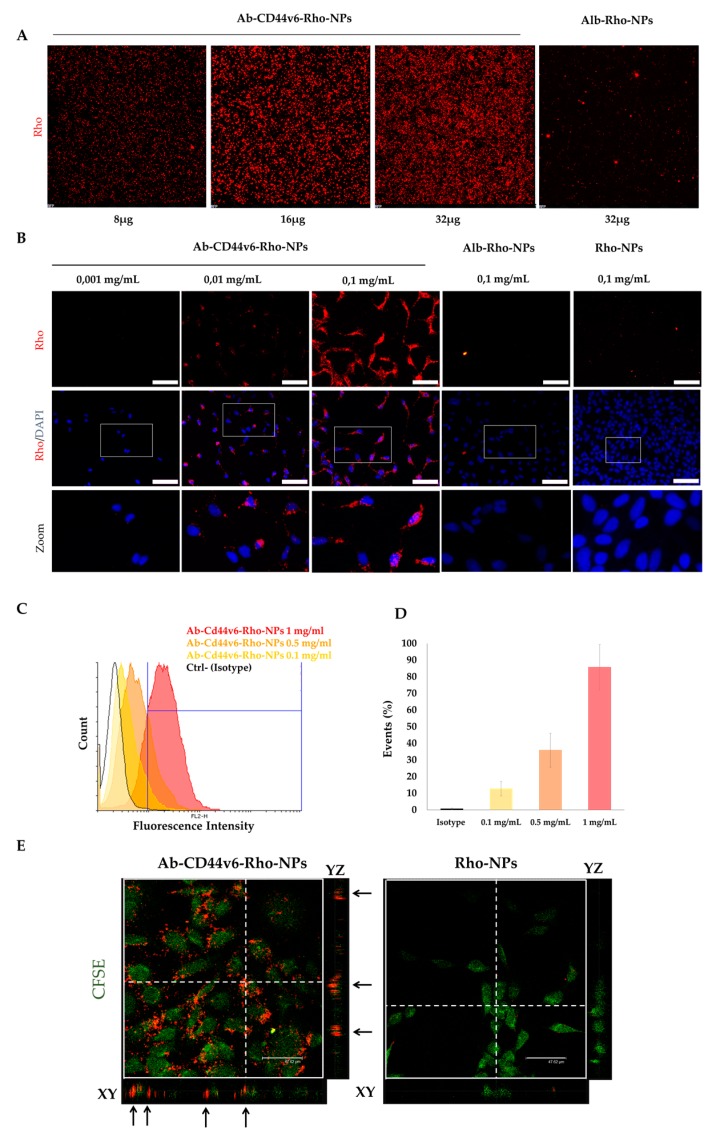
Binding and internalization of Ab-CD44v6-Rho-NPs in HEK-293A-CD44v6: (**A**) Concentration-dependent binding of Ab-CD44v6-Rho-NPs to HEK-293A-CD44v6. Cells were incubated with NPs for 1 h at room temperature (RT) and then washed multiple times to eliminate unbounded NPs; Rho signal (in red); magnification 10×. (**B)** Up-take of different concentrations of Ab-CD44v6-Rho-NPs and of controls, human Albumin conjugated NPs (Alb-Rho-NPs), and unconjugated NPs (Rho-NPs) in HEK-293A-CD44v6; cells were incubated with NPs for 4 h at 37 °C and then washed multiple times to eliminate excess of NPs; DAPI (4′,6-diamidino-2-phenylindole) nuclear staining shown in blue, Rho signal in red; scale bars: 100 µm. (**C**,**D**) Analysis of cellular uptake of Ab-CD44v6-Rho-NPs in HEK-293A-CD44v6 by fluorescence-activated cells sorting (FACS). Cells were incubated with NPs for 4 h at 37 °C, washed multiple times, and fixed. Overlay of histograms (**C**) representative of different experiments performed with the Isotype (Rho-NPs) and different concentrations of Ab-CD44v6-Rho-NPs and (**D**) mean percentage of internalized NPs in HEK-293A-CD44v6: the results are shown for each concentration condition as mean percentage ±SD of experiments performed in triplicates. (**E**) Z-stack analysis of the internalization of Ab-CD44v6-Rho-NPs, compared to unconjugated Rho-NPs, in HEK-293A-CD44v6. A CFSE cell tracer was used to stain the cytoplasm (in green). Dotted lines represent the section in the axial plane corresponding to the frontal (XY) and sagittal (YZ) planes. Black arrows show Rho signal in correspondence to the CSFE signal in frontal and sagittal planes. Scale bars: 47.62 µm.

**Figure 9 nanomaterials-10-00298-f009:**
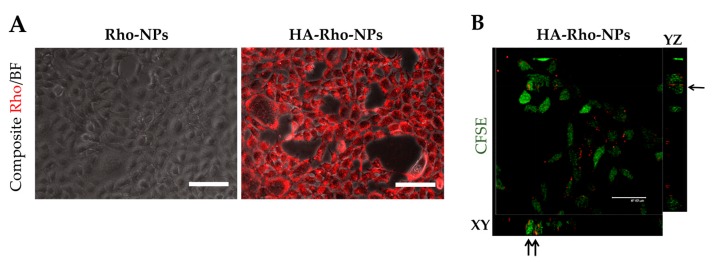
Internalization of HA-Rho-NPs in HEK-293A-CD44v6. (**A**) Internalization of NPs loaded with Rho and conjugated with 22.5 kDa HA (22.5 kDa HA-Rho-NPs) into HEK-293A-CD44v6.; cells were incubated with NPs for 4 h at 37 °C and then washed multiple times to eliminate excess of NPs; Rho signal in red, bright field (BF) in gray; scale bars: 100 µm. (**B**) Z-stack analysis of the internalization of HA-Rho-NPs (0.1 mg/mL) in HEK-293A-CD44v6. A CFSE cell tracer was used to stain the cytoplasm of cells (in green). XY and YZ correspond to the frontal and sagittal planes, respectively. Black arrows show Rho signal in correspondence to the CFSE signal in frontal and sagittal planes. Scale bars: 47.62 µm.

**Figure 10 nanomaterials-10-00298-f010:**
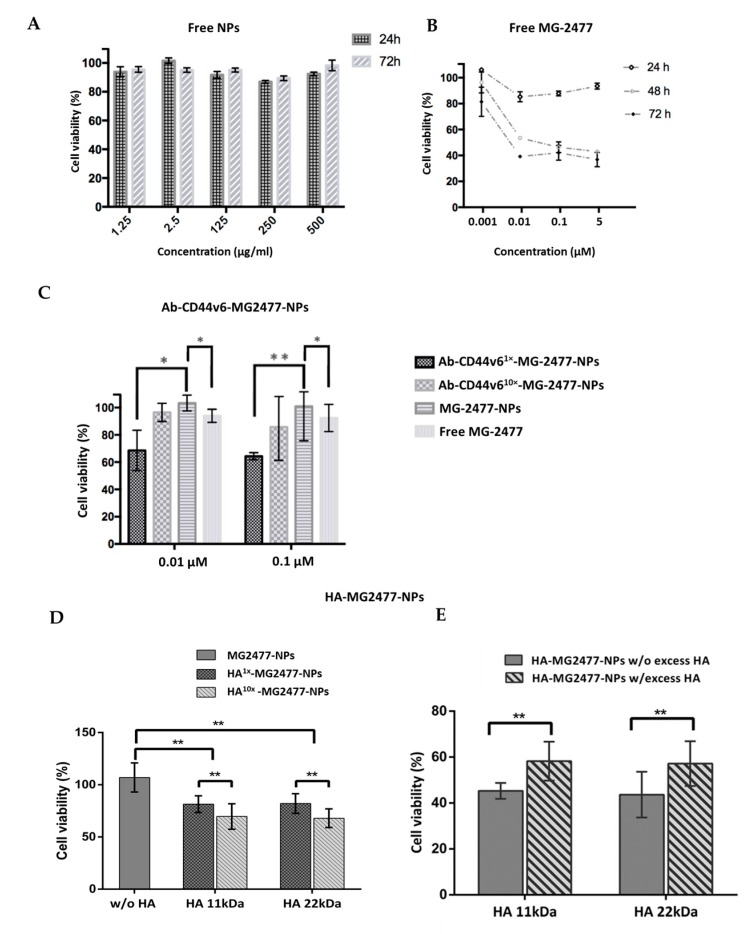
Cytotoxicity analysis of Ab-CD44v6-MG2477-NPs and HA-MG2477-NPs in HeLa-CD44v6 cells. (**A**) Analysis of the cytotoxicity of unconjugated and unloaded NPs (free NPs) by MTT assay. Unconjugated and unloaded NPs were incubated with cells at 37 °C for 4 h, after which the cells were washed to eliminate non-internalized NPs. At 24 and 72 h post-treatment, MTT assay was performed to measure cell viability. (**B**) Analysis of the cytotoxicity of free MG2477 by MTT assay. Cells were treated with free MG2477 for 24, 48, and 72 h, after which MTT assay was performed to measure cell viability. (**C**) Analysis of the cytotoxicity of Ab-CD44v6-MG2477-NPs loaded with MG-2477 and used to treat cells at two different concentration of drug (0.01 and 0.1 µM). NPs Ab-CD44v6^10x^-MG2477-NPS and Ab-CD44v6^1x^-MG2477-NPs, conjugated with a greater or a lower amount of antibody (ratio of NPs/antibody 1:10 and 1:1, respectively) were incubated with cells at 37 °C for 4 h, after which cells were washed to eliminate non-internalized NPs. At 72 h post-treatment, MTT assay was performed to measure cell viability; * *p* ≤ 0.05; ** *p* ≤ 0.01. (**D**) Analysis of the cytotoxicity of HA-MG2477-NPs, conjugated to two different molecular weights of HA (11.5 and 22.5 kDa) and to two different amount of HA (ratio of NPs/HA 1:10 and 1:1, respectively, HA^10x^-MG2477-NPs and HA^1x^-MG2477-NPs,), and used at a drug concentration of 0.1 µM. NPs were incubated with cells at 37 °C for 4h, after which cells were washed to eliminate non-internalized NPs. At 72 h post-treatment, MTT assay was performed; ** *p* < 0.01. (**E**) Competition assay of HA-MG2477-NPs conjugated to two different molecular weights of HA (11.5 and 22.5 kDa) used at a drug concentration of 0.1 µM and pretreated with an excess of HA. Cells were incubated for 1 h at 37 °C with (w/) or without (w/o) excess of HA (10 mg/mL in DMEM medium), washed, and then incubated with for 4 h. At 72 h post-treatment, cells were analyzed by the MTT assay; ** *p* < 0.01. The graphs shown represent the mean percentage of cell survival ±SD of experiments performed in triplicates. For statistical analysis, a Student’s T test was applied.

**Table 1 nanomaterials-10-00298-t001:** Hydrodynamic diameter and Z potential values for the non-conjugated and conjugated nanoparticles.

Entry	Nanoparticles	Size ^a^ (nm)	PDI	Z-Potential ^a^ (mV)
1	NH_2_-NPs	96.0 ± 2.34	0.019	−1.00 ± 0.63
2	Ab-CD44v6^1x^-NPs	104 ± 17.3	0.539	−3.62 ± 0.35
3	Ab-CD44v6^10x^-NPs	99.0 ± 18.9	0.504	−1.60 ± 0.52
4	NH_2_-NPs	140 ± 0.83	0.095	−1.32 ± 0.39
5	11.5 kDa HA^1x^-NPs	144 ± 2.68	0.034	−7.07 ± 0.76
6	11.5 kDa HA^10x^-NPs	137 ± 2.47	0.050	−8.50 ± 0.15
7	22.5 kDa HA^1x^-NPs	128 ± 9.17	0.096	−8.06 ± 0.54
8	22.5 kDa HA^10x^-NPs	127 ± 3.04	0.076	−8.73 ± 0.28

^a^ Data obtained by DLS.

**Table 2 nanomaterials-10-00298-t002:** Loading of anti-cancer drug into unconjugated and conjugated NPs.

Entry	Nanoparticles	Drug Loading ^a^ (w/w %)
1	NH_2_-MG2477-NPs	0.4
2	Ab-CD44v6^1x^-MG2477-NPs	0.05
3	Ab-CD44v6^10x^-MG2477-NPs	0.05
4	11.5 kDa HA^1x^-MG2477-NPs	0.06
5	11.5 kDa HA^10x^-MG2477-NPs	0.07
6	22.5 kDa HA^1x^-MG2477-NPs	0.05
7	22.5 kDa HA^10x^-MG2477-NPs	0.06

^a^ Data obtained by fluorescence emission spectroscopy.

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
