# Peer review of "Multifunctional, CD44v6-Targeted ORMOSIL Nanoparticles Enhance Drugs Toxicity in Cancer Cells"

_nanomaterials, 2020, doi:10.3390/nano10020298_

Round 1
Reviewer 1 Report
Major points
The study focused on formulation of anti-cancer carriers with targeting molecules. The design of the study is okay, but lacks novelty and innovative discovery. In addition, typical cell level toxicity evaluation of drug carriers is not something exciting especially limited number of groups and variables. There is no in vivo results and the system doesn't seem very promissing in vivo. Conventional micelles get solublized instantly by extreme dilutions when administrated. Without a particle stability, the acitive cancer targeting approach with antibody can not be feasible because particles have to travel to the cancer area by passive accumulation prior to antibody-mediated endocytosis.
Minor points
Need to give full name for ORMOSIL when fist used. If the size and zeta-potential measurements were performed with statistically satisfactory number of samples per group, please give ± errors or standard deviation. PDI for ab-conjugated particles is very high (0.5), meaning size of particles measured by DLS is not reliable. There is a chance of particle aggregation due to ab conjugation or that unbound ab in the solution is not completely purified. If the ab to particle ratio based on mass, it would be more useful to know number of ab per single particle, especially due to the fact that 1:10 particle did not work at all. To convince readers, more titration of ab on the particles need to be investigated. For the cytotoxicity study, all formulation including free drug should be equvalent in the amount of drug. If so, this should be discribed in the method.Author Response
(Replies in italic)
The study focused on formulation of anti-cancer carriers with targeting molecules. The design of the study is okay, but lacks novelty and innovative discovery. In addition, typical cell level toxicity evaluation of drug carriers is not something exciting especially limited number of groups and variables.
We agree with the reviewer that we do not have results in vivo. This study was indeed aimed to understand whether is possible to design multifunctional nanoparticles capable to target the CD44 receptor, deliver a drug with solubility issues and increase its activity. For these reasons we preferred to perform a detailed in vitro characterization in order to get precise information on the properties of the systems studied. With this respect we think that the approach is novel and opens the way to further studies.
There is no in vivo results and the system doesn't seem very promissing in vivo. Conventional micelles get solublized instantly by extreme dilutions when administrated. Without a particle stability, the acitive cancer targeting approach with antibody can not be feasible because particles have to travel to the cancer area by passive accumulation prior to antibody-mediated endocytosis.
There was likely a misunderstanding by the reviewer, indeed we are not using micelles but ORMOSIL nanoparticles. Theses nanoparticles do not solubilize by dilution and are remarkably stable in biological fluids. This is well demonstrated by the cell uptake experiments with the MG2477-loaded nanoparticles. Indeed, in the case the nanoparticle dissolved after dilution in culture media, results obtained using the free drug would have been similar to those obtained with the nanoparticles, and targeted and untargeted nanoparticles would have resulted in similar effects.
Need to give full name for ORMOSIL when fist used.
We thank the Reviewer for the suggestion. In the text the full name for ORMOSIL was given when first used at p.3, however it is true that the full name was missing in the abstract. We added the full name also there.
If the size and zeta-potential measurements were performed with statistically satisfactory number of samples per group, please give ± errors or standard deviation.
We added the requested data in table 1.
PDI for ab-conjugated particles is very high (0.5), meaning size of particles measured by DLS is not reliable. There is a chance of particle aggregation due to ab conjugation or that unbound ab in the solution is not completely purified.
The problem of the large PDI of Ab-CD44v6-NPs was discussed in the text (p 12) and indeed attributed to the tendency of the nanoparticles to partially aggregate. Still this does not mean that the size is not reliable: PDI is calculated by the instrument using a mono-modal distribution, while size is calculated using a multimodal distribution. Accordingly, large PDI may indicate the presence of small amounts of large particles or aggregates while the size measured for the main population remains reliable. That this is the case is confirmed by TEM experiments reported in figure 5 that show nanoparticles of the same size provided by DLS (see also figures S3-S4). Purification was checked by SDS-PAGE analysis (Figure 6).
If the ab to particle ratio based on mass, it would be more useful to know number of ab per single particle, especially due to the fact that 1:10 particle did not work at all. To convince readers, more titration of ab on the particles need to be investigated.
These data were indeed provided for the 1x NPs but we missed to report the number for the 10x NPs, we thank the reviewer for noticing it and we added the number.
For the cytotoxicity study, all formulation including free drug should be equvalent in the amount of drug. If so, this should be discribed in the method.
All formulations were used at the same drug concentration (0.1 µM) unless otherwise stated. We changed the text to underline this on the Material and Methods section and on the Results section.
Reviewer 2 Report
The manuscript describes the synthesis and in vitro evaluation of Ormosil nanoparticles decorated with anti-CD44v6 and HA to target tumoral cells. The concept of this system is not really novel because there are a large number of ormosil targeted nanoparticles described with similar aim. However, the results achieved by this system are interesting and it could be published in this journal after addressing some major concerns:
1) Amino groups at the end of the PEG chains are responsible for antibody attachment. These groups are located on the particle surface by the previous synthesis of silane-PEG-NH2 derivative. This compound was synthesized through the reaction with bis-amino-PEG compound and 2-(4-chlorosulfonylphenyl) ethyltrimethoxysilane in equimolar conditions. This proccedure lacks of control and it could yield a mixture of monofunctionalized and bifunctionalized PEG derivatives with silane ends. The authors did not carry out any purification step. Did the authors quantified the amount of amino groups on the particle surface? This data is important in order to achieve an efficient antibody grafting. This amount looks low taking into consideration the negative surface (-1 mV) by Zeta potential measurements.
2) Regarding in vitro interanlization assays, Figure 8 D shows that the amount of NP uptake follows a dose-response behavior increasing the % of events with higher amount of NP, as it is common. However, it is required to use large amount of particles (1 mg/mL) to achieve significant uptake. What is the cell viability with this high nanoparticle concentration?
3) Regarding cytotoxicity assays, the authors should provide any explanation about why nanoparticles decorated with higher amount of antibodies yield almost the same cytotoxicity than non-targeted nanoparticles at 0.01 uM. Additionally, in the case of nanoparticles decorated with antibody (1x) there are not differences when the concentration is increased to 0.1 uM, which is also surprising. The authors should discuss with more details these results. The authors mentioned something in the conclusion section but it only explains the reasons about why it works at low antibody concentration but not why did not work with higher antibody amounts on Np surface.
Author Response
(Replies in italic)
...1) Amino groups at the end of the PEG chains are responsible for antibody attachment. These groups are located on the particle surface by the previous synthesis of silane-PEG-NH2 derivative. This compound was synthesized through the reaction with bis-amino-PEG compound and 2-(4-chlorosulfonylphenyl) ethyltrimethoxysilane in equimolar conditions. This proccedure lacks of control and it could yield a mixture of monofunctionalized and bifunctionalized PEG derivatives with silane ends. The authors did not carry out any purification step. Did the authors quantified the amount of amino groups on the particle surface? This data is important in order to achieve an efficient antibody grafting. This amount looks low taking into consideration the negative surface (-1 mV) by Zeta potential measurements.
We thank the reviewer for pointing out this issue. We did not discuss in detail this point as it was already analyzed in our previous papers on targeting of ORMOSIL nanoparticles with antibodies (refs 24 and 25). It is correct that bifunctionalized PEG may form, still enough amino groups are introduced to allow functionalization with targeting agents, as demonstrated in particular in the case of Ab conjugation, where amino groups introduced revealed to be enough to allow almost full surface coating with antibodies. We modified the text to better explain these points and we added more amine quantification data to table S1.
2) Regarding in vitro interanlization assays, Figure 8 D shows that the amount of NP uptake follows a dose-response behavior increasing the % of events with higher amount of NP, as it is common. However, it is required to use large amount of particles (1 mg/mL) to achieve significant uptake. What is the cell viability with this high nanoparticle concentration?
In our experience the flow cytometer is less sensitive to fluorescent dyes. This is why we used higher concentration of NPs. We have used the same concentrations also to observe the internalization via fluorescence microscopy, that were not reported in the manuscript. In both cases there was no toxicity whatsoever in cells.
3) Regarding cytotoxicity assays, the authors should provide any explanation about why nanoparticles decorated with higher amount of antibodies yield almost the same cytotoxicity than non-targeted nanoparticles at 0.01 uM.
This is an important point. In fact at both concentrations of drug (0.01 uM and 0.1 uM), NPs conjugated with a higher amount of antibody (Ab-CD44v610×-MG2477-NPs, antibody amount 10×) produced a lower cytotoxic effect compared to those conjugated with a lower amount (Ab-CD44v61×-MG2477-NPs, antibody amount 1×) and very comparable to unconjugated MG-2477-NPs. As we stated in the original manuscript, this effect might likely arise form steric hindrance between the nanoparticle grafted antibody molecules. Indeed, in the 10x nanoparticles, antibody quantification indicates that the nanoparticle surface is almost completely covered by antibodies. It is possible that in these highly crowded conditions accessibility of the antibody binding sites is reduced by the presence of the neighboring antibodies molecules. Similar effects have been already described in literature (ref 34 and 35). We modified the text to make this point clearer.
Additionally, in the case of nanoparticles decorated with antibody (1x) there are not differences when the concentration is increased to 0.1 uM, which is also surprising. The authors should discuss with more details these results.
Regarding the NPs conjugated with a lower amount of antibody (Ab-CD44v61×-MG2477-NPs, antibody amount 1×) in both 0.01 uM and 0.1 uM conditions of drug we obtained a similar effect on cell viability reduction. This likely indicates that uptake saturation was already reached at the lower NP concentration. We modified the text accordingly.
Reviewer 3 Report
Nanomaterials-666754
Multifunctional, CD44v6-targeted ORMOSIL nanoparticles enhance drugs toxicity in cancer cells
The manuscript describes drug-loaded PEGylated ORMOSIL nanoparticles prepared by microemulsion condensation of vinyltriethoxysilane. Preparation of amine-functionalized PEGylated ORMOSIL nanoparticles is good and complete. The paper is well written and results based on fluorescence, MTT colorimetric assay, NMR and IR spectra together with DLS and zeta potential measurements are convincing. The only negative point for this referee is the absence of the justification of the statement “We set it to 95 mM to obtain NPs with a diameter of about 120-140 nm” (see line 283). On the other hand authors say in line 344 “sedimentation was also visually observed in the samples after prolonged conservation at 4 °C. TEM images confirmed however that the ORMOSIL cores remained unaffected and that aggregation can be reduced by sonication”. I can not see what advantage the sonication has if in a cellular medium the nanoparticles are going to end up forming aggregates. I advise the authors to justify these two points. Finally English must be revised. It is unacceptable the presence of errors in a manuscript that wants to be published with the quality required by Nanomaterials (MDPI), see for example line 509 or line 24.
Minor points.
1) References 3, 5, 6, 7 can be substituted or extended by more modern ones.
2) Figure 4 is not correct from the point of view of a balance of matter. Compounds are missing in the reaction scheme for the adjustment of matter in the chemical reaction.
3) Figure 10 lacks sufficient clarity and quality and must be modified
Author Response
(replies in italic)
...The only negative point for this referee is the absence of the justification of the statement “We set it to 95 mM to obtain NPs with a diameter of about 120-140 nm” (see line 283).
We thank the reviewer for pointing out this issue. The justification is based on our previous investigations mentioned in the previous sentence. We modified the text to made clearer this point.
On the other hand authors say in line 344 “sedimentation was also visually observed in the samples after prolonged conservation at 4 °C. TEM images confirmed however that the ORMOSIL cores remained unaffected and that aggregation can be reduced by sonication”. I can not see what advantage the sonication has if in a cellular medium the nanoparticles are going to end up forming aggregates. I advise the authors to justify these two points.
The reviewer is correct, the sentence was not correctly formulated. We just wanted to say that the tendency to aggregation, which was indicated by the PDI increase, was only marginal. We rephrased the text accordingly.
Finally English must be revised. It is unacceptable the presence of errors in a manuscript that wants to be published with the quality required by Nanomaterials (MDPI), see for example line 509 or line 24.
We apologize for the English errors. We carefully revised the whole manuscript.
1) References 3, 5, 6, 7 can be substituted or extended by more modern ones.
We replaced the references indicated by the Reviewer with more recent ones and we also added new references in the introduction section.
2) Figure 4 is not correct from the point of view of a balance of matter. Compounds are missing in the reaction scheme for the adjustment of matter in the chemical reaction.
We corrected and simplified figure 4.
3) Figure 10 lacks sufficient clarity and quality and must be modified.
We thank the Reviewer for the observation. We have modified the figure by changing the graph legends, adding titles and by modifying the figure caption to provide more clarity.
Reviewer 4 Report
In the work entitled “Multifunctional, CD44v6-targeted ORMOSIL nanoparticles enhance drugs toxicity in cancer cells”, Lucia Morillas Becerril and co-workers propose the use of MG2477 PeGylated ORMOSIL nanoparticles targeting CD44 receptor expressed in CSCs, to inhibit tumour progression.
In general, the manuscript is poorly written, experiments are poorly designed, and the obtained results were not convincing, making the conclusions doubtful.
Introduction section fail to correctly introduce the state of the art of the work. Specifically, the introduction to the “cancer” topic is not completely correct, several phrases require references and are inaccurate. One example is the phrase in lines 40-42: “Passive targeting, however, relies on solid tumors and on the development of specific physiological features (i.e. immature vascularization), and may not reliably produce the expected benefits.” This phrase is not completely correct as the passive targeting of nanoparticles also occurs due to acidification of the TME. Another example is the introduction to CSCs (lines 47-49) where authors fail to explain correctly how CSCs are formed in the tumour and why they are involved in tumour progression. One other example is the phrase in line 75, “Silica nanoparticles have been among the most studied nanomaterials for biomedical applications in the last decade”, this statement lacks references or scientific support.
Materials and Methods section is poorly written and incomplete. Some examples are that authors do not present the correct percentage units (% (v/v) or % (w/v)), the centrifugation velocity should be presented as xg and not rpm (unless the used rotor and centrifuge are defined), they do not specify which plasmid DNA was used and do not present the conditions of the RT-PCR and how they stained the cells with CFSE. Moreover, they used incorrect scientific language, some examples include line 232, “the RNA was reverse transcribed”, “4*104 cells/24 well” in line 238, “free DMEM” in line 252. The authors also fail to refer the meaning of the abbreviation the first time is mentioned in the abstract and throughout the text, e.g. “PFA”, “MTS”, or “NPs”.
Results section have major inconsistencies, is poorly written and is incomplete. I will just give a few examples. The standard deviation of the measurements should be presented. Also, it is not scientifically correct to use the expressions “approximately -1 mV” or the phrase in line 321 “analyses confirmed that the average diameter of the 320 nanoparticles before conjugation is between 100 and 140 nm depending on the sample”. Figure 8 is very confusing, the caption does not explain the figure; controls of fluorescence images are missing; it is not perceptible the correlation between both graphs of figure 8D (that should be separated in 8D and 8E, for example); in the z-stack of the confocal image is not perceptible that NPs were internalized, in fact, it seems that image was stretched; the language is not correct: “signal can be appreciated inside cells” is not scientific; the cellular density is different in figure 8A, 8B and 8C; the size of scale bars of figure 8B is not specified.
Importantly, and the major reason why I do not suggest the publication of the manuscript, is that the analysis of results presented in figure 10 suggest that proposed nanoparticles does not induce high cytotoxicity in vitro – Nanoparticles that induce the loss of 60 % of cell viability cannot be considered as good cancer therapeutic agents.
Author Response
(Replies in italic)
Introduction section fail to correctly introduce the state of the art of the work. Specifically, the introduction to the “cancer” topic is not completely correct, several phrases require references and are inaccurate. One example is the phrase in lines 40-42: “Passive targeting, however, relies on solid tumors and on the development of specific physiological features (i.e. immature vascularization), and may not reliably produce the expected benefits.” This phrase is not completely correct as the passive targeting of nanoparticles also occurs due to acidification of the TME.
We thank the reviewer for pointing out this issue. We have completely revised the introduction also adding recent references. In particular we rewrote the description of the "passive and active targeting in tumors’’, introducing the required references.
Another example is the introduction to CSCs (lines 47-49) where authors fail to explain correctly how CSCs are formed in the tumour and why they are involved in tumour progression.
We also modified the introduction section about CSCs by explaining how CSCs are formed in the tumor and why they are involved in tumor progression based on recent scientific works.
One other example is the phrase in line 75, “Silica nanoparticles have been among the most studied nanomaterials for biomedical applications in the last decade”, this statement lacks references or scientific support.
References were added to the section on silica nanoparticles to support the statements made.
Materials and Methods section is poorly written and incomplete. Some examples are that authors do not present the correct percentage units (% (v/v) or % (w/v)), the centrifugation velocity should be presented as xg and not rpm (unless the used rotor and centrifuge are defined), they do not specify which plasmid DNA was used and do not present the conditions of the RT-PCR and how they stained the cells with CFSE. Moreover, they used incorrect scientific language, some examples include line 232, “the RNA was reverse transcribed”, “4*104 cells/24 well” in line 238, “free DMEM” in line 252. The authors also fail to refer the meaning of the abbreviation the first time is mentioned in the abstract and throughout the text, e.g. “PFA”, “MTS”, or “NPs”.
We apologize for errors in the Material and Methods section. We completely rewrote those paragraphs to improve scientific language and add missing information. In particular we added more information about the nanoparticle conjugation, the plasmid use, and we specified RT-PCR conditions. We also added a paragraph for CFSE staining. We revised the whole manuscript to add the meaning of abbreviations not explained.
Results section have major inconsistencies, is poorly written and is incomplete. I will just give a few examples. The standard deviation of the measurements should be presented. Also, it is not scientifically correct to use the expressions “approximately -1 mV” or the phrase in line 321 “analyses confirmed that the average diameter of the 320 nanoparticles before conjugation is between 100 and 140 nm depending on the sample”.
The whole results section was revised also accordingly to the comments of the other reviewers.
Figure 8 is very confusing, the caption does not explain the figure; controls of fluorescence images are missing; it is not perceptible the correlation between both graphs of figure 8D (that should be separated in 8D and 8E, for example); in the z-stack of the confocal image is not perceptible that NPs were internalized, in fact, it seems that image was stretched; the language is not correct: “signal can be appreciated inside cells” is not scientific; the cellular density is different in figure 8A, 8B and 8C; the size of scale bars of figure 8B is not specified.
We thank the Reviewer for the comment. We have modified the caption of Fig 8 and added extra information to increase clarity. We have combined fig. 8B-C (now 8B) to highlight the controls (NPs conjugated with Alb or free NP as negative control). We have added more images in the supplementary materials related to fig. 8A-B. We have separated graphs of Figure 8D (Now 8C-D) and described them in the caption. Images of Z-stack are the original ones but we might have failed in making a clear description: we emphasized the dotted lines in the figure that represent the sections of the sagittal and frontal planes and added the coordinates of the plane together with a broader description in the caption to clarify the image. Scale bars were specified in all images. Cellular density in microscopy images represent different areas of the well taken randomly. Therefore it might be different from image to image.
Importantly, and the major reason why I do not suggest the publication of the manuscript, is that the analysis of results presented in figure 10 suggest that proposed nanoparticles does not induce high cytotoxicity in vitro – Nanoparticles that induce the loss of 60 % of cell viability cannot be considered as good cancer therapeutic agents.
We agree with the Reviewer that a loss of 60% of cell viability cannot be considered as good cancer therapeutic agent. We would like to underline however that this is the maximum reduction of cell viability obtained by the free drug in optimal conditions (more the 48h incubation) and that a similar result was obtained with the drug-loaded nanoparticles in conditions where the free drug is poorly ineffective. Hence while the overall effectiveness of the system is not optimal, the delivery vehicle is working very well. The main goal of our work was indeed to evaluate the potential of our nanoparticle formulations. With a smart synthetic procedure, we can obtain nanoparticles that are able to target an overexpressed cell marker and deliver a drug achieving full effectiveness, in conditions where the same drug is ineffective when delivered without the nanoparticles. This proof of principle study reveals that candidates that were discarded in pre-clinical studies because of their scarce activity or poor solubility could be re-evaluated in targeted nanoparticles formulations. In addition, they show that low cost and easy to prepare targeting agents as HA can be as effective as antibodies. We modified the conclusions to make these points clearer.
Round 2
Reviewer 1 Report
The study lacks novelty and innovative discovery to be published in Nanomaterials.
Author Response
The study lacks novelty and innovative discovery to be published in Nanomaterials.
We regret we could not convince Reviewer 1 of the relevance of the studies here reported. We can only recall that the work received positive comments, with respect to its relevance, by the other three reviewers.
Reviewer 4 Report
The corrections made by authors were in agreement with my previous suggestions. I just have minor questions:
References are missing in lines: 71, 90, 93, 94, 95, 117, 123, 125, 277.
Line 189 – define milliQ water – distilled and deionized water (milliQ water is a trademark)
Line 300: correct the phrase: “For Z-stack analysis, cells were For 300 FACS analysis: internalized evaluated by 301 detaching the cells with trypsin, washing them with PSB and by analyzing the fluorescent signal 302 with a BD FACSCalibur (BD Bioscience, Franklin Lakes, New Jersey, USA).
Author Response
References are missing in lines: 71, 90, 93, 94, 95, 117, 123, 125, 277.
We added all the missing references but not at lines 93-95 (Caption to Figure 1, references not needed since they are in the text where the figure is discussed) and lines 123-125 (summury of the main findings of this paper at the end of the introduction, no reference possible)
Line 189 – define milliQ water – distilled and deionized water (milliQ water is a trademark)
We substituted MilliQ with ultrapure, mentioning this term also where the production of ultapure water is described.
Line 300: correct the phrase: “For Z-stack analysis, cells were For 300 FACS analysis: internalized evaluated by 301 detaching the cells with trypsin, washing them with PSB and by analyzing the fluorescent signal 302 with a BD FACSCalibur (BD Bioscience, Franklin Lakes, New Jersey, USA).
Corrected